# Phylogenetic Analysis of a Newcastle Disease Virus Strain Isolated from Domestic Poultry and Its Potential for Vaccine Development in the Republic of Kazakhstan

**DOI:** 10.3390/vaccines13050440

**Published:** 2025-04-23

**Authors:** Nurlan Akmyrzayev, Sholpan Ryskeldinova, Aigerim Mailybayeva, Yerken Kozhamkulov, Aigerim Sagymbayeva, Yeldos Myrzakhmetov, Yerbol Burashev, Nurlan Kozhabergenov, Bekbolat Usserbayev, Nurika Assanzhanova

**Affiliations:** Research Institute for Biological Safety Problems, National Holding “QazBioPharm”, Gvardeiskiy 080409, Kazakhstan; sh.ryskeldinova@biosafety.kz (S.R.); a.mailybayeva@biosafety.kz (A.M.); e.kozhamkulov@biosafety.kz (Y.K.); a.sagymbayeva@biosafety.kz (A.S.); y.myrzakhmetov@biosafety.kz (Y.M.); y.burashev@biosafety.kz (Y.B.); n.kozhabergenov@biosafety.kz (N.K.); b.usserbayev@biosafety.kz (B.U.)

**Keywords:** Newcastle disease, sequencing, phylogenetic analysis, genotype, vaccine, stability, harmlessness, immunogenicity, efficacy

## Abstract

Background/Objectives: Despite the availability of Newcastle disease (ND) vaccines, outbreaks have continued to occur for more than six decades, with significant economic consequences for the global poultry industry. The variability of the Newcastle disease virus requires constant monitoring, detection of new cases, and studies of the origin of the pathogen. The aim of this study was to develop an inactivated ND vaccine using a topical strain with different adjuvants and to compare them for stability, harmlessness, immunogenicity, and efficacy. Methods: A phylogenetic study of the F-gene of the ND strain isolated in Kazakhstan was conducted. The strain, which was selected to create a vaccine for the prevention of the disease, was revealed to belong to genotype VII class II and uploaded to GenBank (NCBI). Two different adjuvants, Montanide ISA 70 VG and Montanide ISA 78 VG, were used to create the vaccine. Birds were vaccinated intramuscularly. Results: Evaluations of antibody titers in the vaccinated groups during the experiment showed that the vaccines induced adequate levels of antibodies to provide protection against the virulent virus. High antibody titers were observed in the hemagglutination inhibition assay (HAI) in the vaccinated groups as early as 14 days post-vaccination in 100% of birds. The average antibody titer in both vaccinated groups exceeded 7 log2, sufficient to prevent clinical signs. None of the vaccinated birds exhibited clinical signs following control infection, whereas unvaccinated birds developed clinical manifestations within three days post-infection, leading to 100% mortality. Conclusions: The vaccine developed using the epizootic topical strain is stable, harmless, immunogenic, and effective when challenged with a virulent ND virus strain at a dose of 105 EID_50/mL_.

## 1. Introduction

Newcastle disease (ND) is a highly contagious viral disease that mainly affects birds, especially chickens. It is characterized by its rapid coverage of a large number of birds and high mortality, pneumonia, encephalitis, and the manifestation of hemorrhagic syndrome in the form of multiple punctate hemorrhages in internal organs. It causes significant economic damage and constitutes an especially dangerous infection [1]. ND outbreaks can be devastating, and the mortality rate from cycling ND in poultry reaches 100% [2].

According to the ICTV taxonomy, Newcastle disease virus (NDV) belongs to the species *Orthoavulavirus javaense* of the genus *Orthoavulavirus*. It is a negative-sense, single-stranded, enveloped RNA virus. It has an envelope of a negative-polar single-stranded RNA genome approximately 15.2 kb in size, encoding six structural proteins: nucleocapsid protein (NP), phosphoprotein (P), matrix protein (M), fusion protein (F), hemagglutinin–neuraminidase (HN), and large polymerase protein (L). Among these proteins, F is considered the main molecular determinant of ND’s pathogenicity [3,4].

Newcastle disease virus isolates vary in their pathogenicity for chickens and may include at least three pathotypes: lentogenic (low virulence), mesogenic (moderate virulence), and velogenic (high virulence). Velogenic ND remains a serious threat to poultry production due to its high mortality and reduced egg production in laying hens [5]. This classification system plays a key role in the diagnosis and control of disease progression, as well as in the development of effective vaccine strategies [6,7].

For many years, live attenuated vaccines prepared from La Sota, B1 genotype II strains have been used in the global poultry industry to reduce the threat of ND [8,9]. A single injection of 10^5^ EID_50_ of live ND vaccine, which can confer 100% protection against clinical disease, is sufficient to quickly stimulate the immune response. Despite all these advantages of live attenuated vaccines, they also have their drawbacks. First, it is worth noting that live viruses are able to restore virulence and cause disease outbreaks even in vaccinated birds. [10]. In addition, these vaccines can cause respiratory reactions in young birds, which, if severe, can lead to secondary bacterial infections. [11] Moreover, these strains, developed in the 1940s, are 21–23% genetically different from the viruses currently circulating in different parts of the world [12]. Significant antigenic and genetic diversity was observed between different ND genotypes, despite the fact that they are all APMV-1 serotypes. Currently, the most common ND genotype circulating in Asia [13,14], the Middle East [15], South Africa [16], South America [9], and Europe [17] is genotype VII, which is associated with the ongoing fourth disease pandemic that began in the late 1980s [18].

Given the existing problem of genotype mismatch and the associated decrease in the effectiveness of the vaccines used, the aim of this work was to develop a safer and more effective inactivated Newcastle disease vaccine. These vaccines should be available to farmers and be able to effectively prevent the development of the disease in birds. Several recent studies have shown that inactivated vaccines developed from currently circulating genotype strains provide better protection than traditional vaccines [19].

Since 2021, the Research Institute for Biological Safety Problems (RIBSP) has been conducting research on the development of inactivated Newcastle disease vaccines within the framework of PCF O.001B, resulting in the development of a safe and effective inactivated vaccine based on the recently circulating ND strain.

Inactivated vaccines used to protect poultry require the selection of strains and adjuvants in order to induce a sufficient level of immune response. The safety of the adjuvant in the emulsion of the water–oil composition is important for the development of an effective inactivated Newcastle disease vaccine [20]. The use of the ready-to-use Montanide ISA system as an adjuvant reduces the number of technological steps in the preparation of the emulsion, thereby significantly distinguishing the overall duration of vaccine preparation compared to the use of multicomponent adjuvants that require additional time and money and the cost of preparing emulsions [21].

## 2. Materials and Methods

### 2.1. Sequencing the ND Virus F-Gene

This study used the Newcastle disease virus strain PMV-1/Astana/chicken/49/98. This strain was isolated from domestic poultry in Kazakhstan by the Institute of Virology and Microbiology (Almaty, Kazakhstan) and maintained in the microbial collection laboratory of the Research Institute for Biological Safety Problems (RIBSP), Kazakhstan.

A pre-prepared viral suspension obtained from SPF eggs sourced from the Podmoklov nursery (Moscow, Russia) was used. Viral RNA extraction was performed using a validated QIAamp Viral RNA Mini Kit (Qiagen GmbH, Hilden, Germany), according to the manufacturer’s instructions.

Reverse transcription was performed using the UltraScript cDNA Synthesis Kit (PCR Biosystems Ltd., London, UK) [22]. The amplification of the F-gene was performed using the VeriFi Mix kit (PCR Biosystems Ltd., London, UK) and sequencing primers, the sequence of which is presented in Table 1.

The PCR purification of the products was carried out using the innuPREP DOUBLEpure Kit (Analytik Jena GmbH + Co. KG, Jena, Germany), according to the manufacturer’s instructions.

The ND virus F-gene sequencing was performed using dideoxynucleotides, according to the Sanger method, using the BigDye Terminator v3.1 Cycle Sequencing Kit from Thermo Fisher Scientific (Waltham, MA, USA) and with overlapping primers (Table 1) used during the preparation phase. The resulting products were purified using the BigDye Xterminator kit (Thermo Fisher Scientific, Waltham, MA, USA) and then sequenced using the 3130XL genetic analyzer from Applied Biosystems (Thermo Fisher Scientific, Waltham, Massachusetts, USA) and Hitachi Ltd. (Tokyo, Japan). After the sequencing was completed, the nucleotide sequence data were analyzed using the Sequencher v.5.4 software developed by Gene Codes Corporation (Ann Arbor, MI, USA).

### 2.2. Phylogenetic Analysis of ND Virus F-Gene

Phylogenetic analysis based on the F protein gene was performed using the MEGA 11 program and the adjacent conjunction method with 1000 bootstrapping repeats [3]. The sequences used for the phylogenetic analysis were derived from various genotypes of the ND strain registered with GeneBank (NCBI).

The evolutionary history was constructed using the neighbor-joining method [23], and the optimal phylogenetic tree was constructed. Below the branches, the percentages of duplicate trees in which related taxa were grouped as part of a bootstrap test (1000 iterations) are indicated [24]. The tree is depicted to scale, with the length of the branches measured in the same units as the evolutionary distances that were used to build the tree. These distances were calculated using the maximum likelihood method [25] and are expressed as the number of nucleotide substitutions per site. In total, 59 nucleotide sequences were involved in the analysis. All undefined positions were eliminated for each pair of sequences.

### 2.3. Inactivation and Formulation of Vaccines

For the preparation of experimental batches of vaccines, the strain of the Newcastle disease virus “PMV-1/Astana/chicken/49/98” was used; it was produced by inoculating 10–11-day-old chicken embryos into the allantoic cavity. The level of the hemagglutinating activity of the viral strains was determined based on the hemagglutination reaction, conducting a series of double dilutions of the material in 96-well plates, after which a 1% suspension of rooster erythrocytes was added [1].

The virus was inactivated using a formaldehyde solution with a concentration of 0.05%. To analyze the residual virulence of the inactivated material, three consecutive blind passages were performed in 10-day-old chicken embryos by inoculation into the allantoic cavity, according to the technique described in [26].

When developing emulsion vaccines of the “water-in-oil” type using oil adjuvants of the Montanide ISA 70 VG (vaccine-1) and Montanide ISA 78 VG (vaccine-2) series (Seppic, France), we followed the manufacturer’s instructions. The vaccine was prepared in the form of a water-in-oil (W/O) emulsion. The aqueous phase comprised 30% inactivated material of the corresponding strain, while the oil phase constituted 70% and contained the adjuvant Montanide™ ISA70 or 78 VG. To obtain a homogeneous W/O emulsion, the aqueous phase was gradually introduced into the oil phase with thorough homogenization at 2500 rpm for 30 min. The antigen was prepared with hemagglutinating activity at a level of 1:64. All procedures were performed according to [20,27].

The quality of the experimental batch of the inactivated Newcastle disease vaccine was evaluated according to the following criteria: sterility, safety, pH level, kinetic viscosity, emulsion stability, and immunogenic efficacy.

The stability of the emulsion was studied by testing its long-term storage at temperatures of 4 °C and 25 °C, according to the method given in [28].

### 2.4. Harmlessness Tests of Inactivated Newcastle Disease Vaccines

The studies were carried out in specialized premises that fully met the relevant sanitary and hygienic requirements. The temperature in the room ranged from 20 to 24 °C, the humidity level did not exceed 50%, and the lighting followed a day–night mode. Throughout the period of monitoring the immunogenic efficacy and safety, the chicks received adequate nutrition and had unlimited access to water and feed. All experiments with animals were conducted in accordance with the principles of bioethics that govern the practices of working with laboratory animals. To assess the safety and immunogenicity of the vaccines, 28-day-old Super Nick breed chickens were used. These chickens were hatched from SPF eggs obtained from the Podmoklov nursery.

To assess the safety of the vaccines, we used 20 chickens at the age of 28 days. The chickens were randomly divided into two groups: G1, which received the inactivated Vaccine-1 vaccine (n = 10), and G2, which received the inactivated Vaccine-2 vaccine (n = 10). The vaccine was considered safe if, within 10 days of the administration of the vaccine in a 5-fold dose, the vaccinated chickens did not experience clinical symptoms of the disease (such as depression, a loss of sensitivity, cyanosis of visible mucous membranes, etc.) [29] or cases of mortality.

The local tissue reaction at the site of vaccination in the chickens was assessed on the 10th day after vaccination. To this end, the animals were taken out of the experiment by being given an overdose of Propapol (0.5 mL intravenously), after which an autopsy was performed. The condition of the muscles in the injection area was analyzed and the degree of tissue reaction was determined according to the following gradation: moderate (pallor of the muscles in the injection area and the absence of vaccine residues), moderate (the muscles were pale or red, and vaccine residues were present), and severe (the inflammation of superficial and deep muscles and vaccine residues at the injection sites) [30].

### 2.5. Determination of Vaccine Immunogenicity

Before use, the vaccine vials were kept at a temperature of 18–20 °C for 8–10 h. Prior to vaccination, blood serum was taken from the chickens to assess their immune state and it was analyzed simultaneously with the sera obtained 7, 14, 21, and 28 days after vaccination in the hemagglutination inhibition reaction (HAI). A group of 30 chickens that were 28 days old were used to assess the immunogenicity of the vaccines. By randomization, the chickens were divided into three groups: G3—chickens that received Vaccine-1 (n = 10), G4—chickens that received Vaccine-2 (n = 10), and a control group (CG) (n = 10).

### 2.6. A Test for the Efficacy of the Inactivated Newcastle Disease Vaccines After a Single Vaccination

To assess the efficacy of the vaccines studied on day 28, the G3, G4, and CG chickens participating in the immunogenicity experiment were infected with an epizootic virulent strain of the ND virus at a dose of 10^5^ EID_50_/_mL_ with a volume of 0.5 mL injected into the pectoral muscle. For 10 days after viral infection, the chickens were examined daily and assessed on a scoring scale. The assessment of clinical manifestations in birds was as follows: 0 points—no symptoms, 1 point—a loss of appetite or refusal to drink, 2 points—mild respiratory symptoms or greenish diarrhea, 3 points—severe condition before death, and 4 points—death. Mean values were evaluated as the total score for each group. Birds that had the infection and did not show clinical signs were considered protected (Appendix A). According to the monograph of the European Pharmacopoeia, a vaccine is considered effective if the level of actual protection reaches 50% [26].

### 2.7. Statistical Analysis

A statistical analysis of all the experimental data was performed using Graph Pad Prism software version 8.0 (Graph Pad Software Inc., La Jolla, CA, USA). The titer of specific antibodies was expressed as the mean value with standard errors. The statistical difference in antibody titers between intervals was assessed using Student’s test (*t*-test). *p* < 0.05 was considered significant.

### 2.8. Ethics Statement

The study and use of animals in the experiments were carried out with the approval of the Local Ethics Commission on Bioethics of the RIBSP (Protocol No. 1 of 10 January 2021).

## 3. Results

### 3.1. Genetic and Biological Characteristics

As a result of the research, it was possible to successfully extract viral RNA, which became the basis for the subsequent polymerase chain reaction. The analysis of the results using a transilluminator MiniBIS Pro and GelCapture software confirmed the presence of amplified products, which indicates the accuracy of the steps performed and the effectiveness of the primers used for the F-gene of the ND virus.

According to the data (Figure 1), the “PMV-1/Astana/Chicken/49/98” strain is located on the same branch as strains from China (OP818781.1), Japan (AB853927.2), Iran (KU201410.1) and the Russian Federation (OQ190211.1), but is not closely related to widely used vaccine strains, such as Clone 30 and La Sota (AF077761.1). These results allow us to identify possible risks associated with the use of existing vaccines that do not provide adequate protection against new strains. This highlights the importance of monitoring virulent strains, especially in the context of the spread and evolution of viruses.

The complete F-gene sequences of the virulent ND virus reviewed in this study have been uploaded to GenBank (NCBI) and are available at the following access number: PP738481.1.

### 3.2. Phylogenetic Analysis of F Protein

The phylogenetic analysis of the F-gene showed that the strain “PMV-1/Astana/chicken/49/98” belongs to the genotype VII class II, which contained amino acid sequences of RRQKRF, indicating its velogenicity. It is notable that the closest to the “PMV-1/Astana/chicken/49/98” strain was the ch-77/01 isolate from China, with an identity percentage of 98.26%. This may indicate that virulent strains are actively circulating in regions with a high poultry density.

### 3.3. Physical and Chemical Parameters of Vaccines

The physicochemical characteristics of the vaccine offer key information about its composition and properties. These indicators play a critical role in assessing a vaccine’s quality and effectiveness and in ensuring its safety during use. As part of our study, we analyzed the following vaccine parameters: sterility, safety, pH, kinetic viscosity, and stability.

Based on the results of our analysis, the physicochemical characteristics of the emulsified vaccines fully comply with the established requirements. During temperature control, the appearance of a transparent aqueous fraction at the bottom of the test tubes, which are generally used for visual inspections, was not noticed. When stored for more than three months at 25 °C or for 12 months at 4 °C, the vaccines remained stable and unaffected. The kinematic viscosity was 38.62 ± 0.01 and 39.20 ± 0.01, which is within the permissible range (20–150 mm^2^/s). An analysis of the concentration of hydrogen ions showed that the level fully meets the requirements. The measured pH values for the inactivated Vaccine-1 were 7.23 ± 0.00, and for Vaccine-2, they were 7.32 ± 0.00, which indicates the neutral acidity of both vaccines. The culture on the nutrient media did not reveal the growth of bacteria, fungi, or mycoplasmas.

These results confirm the high efficiency of the virus inactivation process and demonstrate the safety and efficacy of vaccines in preventing disease.

### 3.4. Vaccine Safety Assessment

For 10 days, we assessed the safety of vaccines against Newcastle disease based on the strain “PMV-1/Astana/chicken/49/98” with various adjuvants. During the observation period, all birds involved in the study remained alive, and no deviations from physiological norms were detected.

An autopsy of the tissues at the injection site showed that there were no pathological changes in the muscles in the injection area; they remained pale, and no vaccine residues were found.

### 3.5. An Analysis of the Immunogenicity Assessment of Inactivated Emulsified Vaccines

One of the goals of this study was to evaluate the vaccine’s ability to induce an immune response in the body against Newcastle disease. As part of the process, experiments were carried out on birds with a subsequent assessment of their effectiveness. A graphical representation of the results of the study is given in Figure 2.

An analysis of the data presented in Figure 2 suggests that the vaccination of seronegative birds at the beginning of the study in both experimental groups induced a significant immune response as early as 14 days after vaccine administration. In birds that received inactivated Vaccine-1 with adjuvant Montanide ISA 70, the active development of immunity was observed: on day 14, the level of antibodies was 5.17 ± 0.4 log2, increasing to 9.19 ± 0.1 log2 on day 21 and reaching 9.27 ± 0.2 log2 on day 28. Birds treated with inactivated Vaccine-2 with adjuvant Montanide ISA 78 also showed a marked increase in antibody titers: 8.89 ± 0.1 log2 on day 14, 9.49 ± 0.1 log2 on day 21, and 9.68 ± 0.2 log2 on day 28. These results indicate the formation of a strong immune response in vaccinated birds, which confirms the high effectiveness of the inactivated vaccine in protecting against Newcastle disease. In contrast, no antibodies were detected in the control group either before the study or during the assessment of vaccine immunogenicity.

### 3.6. Results of Vaccine Protection After a Single-Dose Vaccination

Subsequently, the protection conferred by the inactivated vaccines was assessed based on the clinical manifestations of the disease, survival, and other indicators according to the point system.

Control infection was carried out using the virulent strain “PMV-1/Astana/chicken/49/98”. In unvaccinated chickens infected with 10^5^ EID_50_/_mL_ of the control strain, characteristic clinical manifestations of the cycling form of Newcastle disease were noted, including depression, a loss of response to external stimuli, and cyanosis of the visible mucous membranes, comb and catkins. These symptoms began to appear on the third day after infection and gradually intensified, leading to the death of all chickens on the sixth day. In the G3 and G4 groups, no symptoms or deaths were reported. The statistical analysis revealed a significant difference (*p* < 0.0001) between the control and experimental groups in terms of clinical manifestations and mortality rates (Figure 3). In this study, all vaccinated birds remained alive until the end of the experiment, demonstrating the high efficacy of inactivated vaccines based on the PMV-1/Astana/chicken/49/98 strain with different adjuvants against Newcastle disease after a single vaccination (Appendix A).

## 4. Discussion

According to the OIE, in 2024, cases of Newcastle disease (ND) were recorded in various countries, including industrialized ones such as Sweden, Poland, Israel, Brazil, Russia, Chinese Taipei, and Botswana. In Kazakhstan, the epizootic situation with ND also remains tense, as outbreaks of the previously registered Newcastle disease continue to be observed in the countries bordering Kazakhstan [31].

To protect against Newcastle disease, poultry farms in Kazakhstan use a unified vaccination schedule, where live vaccines against this infection from the strains “La Sota” and “B1” based on genotype II are actively used.

Despite the use of a live vaccine, an effective technology for the production of an inactivated vaccine against Newcastle in Kazakhstan has not yet been developed. Based on the above context, RIBSP has conducted research on the development of inactivated vaccines against Newcastle disease.

The current focus in the control of virulent ND virus infections is the development of genotype-appropriate vaccines based on currently circulating ND virus isolates [9]. In recent outbreaks in Taiwan and China, the genotype VII virus has been identified as the main causative agent. It is present in different parts of the world [14].

In our study, the ND virus gene was studied in detail. The phylogenetic analysis showed that the strain “PMV-1/Astana/chicken/49/98” belongs to the genotype VII class II and has the amino acid sequence RRQKRF. This means that the studied strain isolated in Kazakhstan is velogenic, and it is not closely related to vaccine strains such as Clone 30 and La Sota. An analysis of the genetic variability of ND strains in different regions demonstrates that new genotypes, such as VII, can cause outbreaks even in vaccinated populations where the above traditional vaccine strains are used, as evidenced by the similarity in the spread of the virus and its pathogenicity. This emphasizes the need to update vaccine strains, taking into account the current viral circulation [13,14,15,16,17].

Inactivated vaccines designed to protect birds require the careful selection of not only strains but also adjuvants to ensure the necessary level of immune response. To create an effective vaccine, high-quality antigens are important, but so are safe adjuvants that will help increase the immunogenicity of the antigen [21]. Based on our experience in developing inactivated vaccine manufacturing technologies and reviewing the available literature related to different immunostimulants, we compared vaccines using the same antigen and different oil adjuvants, such as Montanide ISA 70 and Montanide ISA 78. These adjuvants are widely used in the production of vaccines for birds and have the advantage of reducing the number of technological steps in their creation compared to multicomponent adjuvants, which are time- and resource-intensive [20].

In this study, we compared two water-in-oil vaccines from the PMV-1/Astana/Chicken/49/98 strain with the above adjuvants (manufactured by Seppic, France) against Newcastle disease with a study of their safety, immunogenic properties, and efficacy. The series of vaccines against Newcastle disease prepared by us were sterile and did not have foreign impurities; they were safe for the vaccination of chickens, did not cause manifestations of the disease after their introduction into the bodies of chickens, and met the recommendations of the OIE [1]. It was found that the kinematic viscosity of the studied vaccine batches had an acceptable level according to the criterion established as 150 mm2/s, which contributed to the fluidity of the vaccine and ensured its easy passage through the needle during injection [32].

The vaccines maintained the stability of the emulsion at a storage temperature of 4 °C for 12 months in all samples studied. At 25 °C, the duration of stability of the emulsified vaccines was determined to be more than three months; a slight separation of the oil and water phases was observed at week four of follow-up, which easily recovered after intensive shaking even after three months of storage, turning into a homogeneous emulsion. This is permissible according to the manufacturer’s recommendations (Seppic, France) [20,28].

To study the safety of these vaccines, the vaccinated chickens were monitored for 10 days after vaccination. In vaccines, the adjuvant is used to enhance the humoral and cellular immune response, but the prolonged exposure of mineral oil to the injection site, which causes inflammation and local tissue necrosis, can lead to a decrease in the marketable value of poultry [33]. To assess macroscopic tissue injuries in the injection area, an autopsy was performed on day 10, which did not reveal damage or other inflammatory phenomena in the tissues; this finding is confirmed by the results of other studies [31].

To assess the efficacy of the inactivated vaccines with different adjuvants in the formation of a sustained immune response in birds against Newcastle disease, antibody titers were compared in both experimental groups. They showed that, as early as 14 days after vaccination, there was a significant increase in antibody levels, which confirms the activation of the immune system; meanwhile, the group that received the vaccine with Montanide ISA 78 showed higher antibody titers at all stages of follow-up, starting from day 14, compared to birds treated with Vaccine-1 with Montanide ISA 70. However, over time, antibody levels in the Vaccine-1 group increased significantly, demonstrating that the immune response to the inactivated Montanide ISA 70 adjuvanted vaccine not only begins after 14 days but continues to evolve, peaking at day 28.

It is interesting to note that, despite the high titers of antibodies, there were lower levels of growth dynamics in the Vaccine-2 group compared to the first vaccine. This may indicate differences in the mechanisms of action of the adjuvants and their effect on the formation of immune memory. To compare the immunogenicity and protective efficacy of these vaccines, the responses of avian antibodies in HAI were compared with the results of the control infection. From the literature [32], it is known that an antibody titer below 1:16 (on average > 4 log2) does not protect against infection with wild viruses. The evaluation of antibody titers (HAIs) in the vaccinated groups during the experiment shows that the vaccines were able to induce adequate levels of antibodies to provide protection against the virulent virus. In our experiment, vaccination provided high levels of antibody titers in the HAI in the vaccinated groups G3 and G4; as early as 14 days after vaccination, in 100% of birds, the antibody SHT reached a protective level (on average > 4 log2). On the day of infection, the mean antibody titer in the HAI in both vaccinated groups was above 7 log2, which is more than enough to protect birds from clinical signs. None of the vaccinated OG3 birds showed clinical signs after control infection (0 points). In the unvaccinated birds of the CG, clinical manifestations were observed as early as three days after infection with increasing severity, and eventually, all of these birds died (100% mortality). These data are confirmed by Alexander et al. (2003), who showed that virulent strains can cause high mortality in unvaccinated susceptible herds [34].

## 5. Conclusions

In conclusion, the results of this study highlight the importance of the continuous monitoring and adaptation of vaccination strategies to reduce economic losses in poultry production and control the spread of viral infections. The development of new vaccines based on topical strains, such as genotype VII, is crucial to ensuring bird health and effective protection against Newcastle and ND viruses.

The developed vaccines have demonstrated their effectiveness, but differences in the dynamics of the antibody response suggest that the choice of adjuvant can have a significant impact on the vaccination outcome. More research is needed to enable us to better understand the relevant immunological mechanisms and to optimize vaccinations against Newcastle disease.

## Figures and Tables

**Figure 1 vaccines-13-00440-f001:**
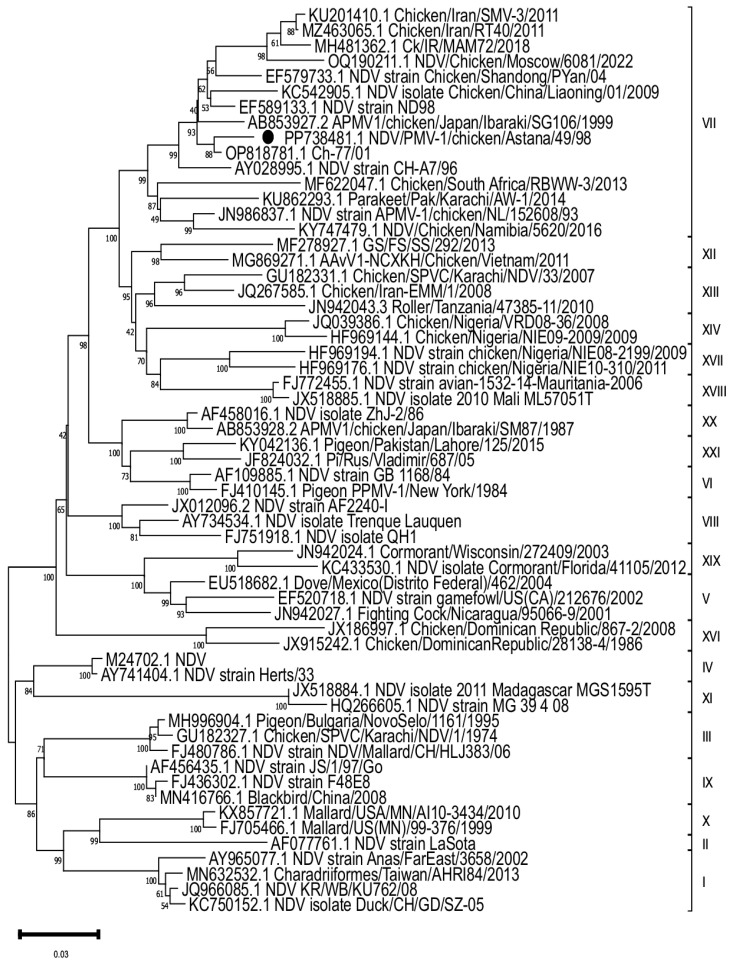
A phylogenetic analysis of the F-gene of the strain “PMV-1/Astana/chicken/49/98”.

**Figure 2 vaccines-13-00440-f002:**
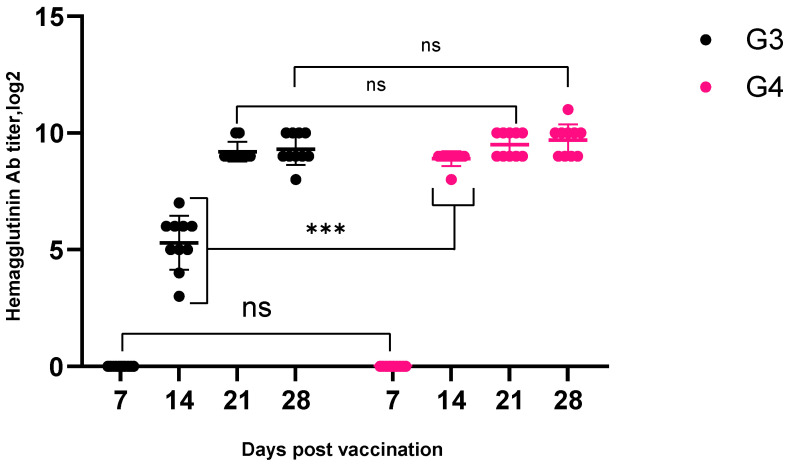
An analysis of the immunogenicity assessment of inactivated emulsified vaccines against Newcastle disease from the strain “PMV-1/Astana/chicken/49/98” with different adjuvants. Antibody titer in the blood of vaccinated chickens from 14 to 28 days after vaccination (log2). Data are means ± standard errors; *** indicates *p* < 0.0005, and ns indicates no significant difference (*p* > 0.05).

**Figure 3 vaccines-13-00440-f003:**
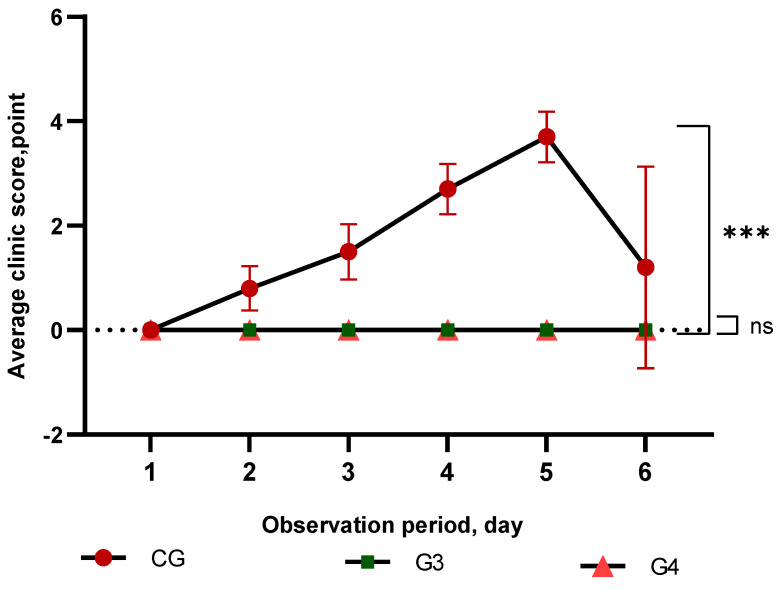
A comparative analysis of the severity of clinical signs in the study of the protectivity of vaccines from the strain “PMV-1/Astana/chicken/49/98” of the ND virus with different adjuvants in experimental and control birds after infection with the control strain. Data are means ± standard errors; indicates *** (*p* < 0.0001, and ns indicates no significant difference (*p* > 0.05).

**Table 1 vaccines-13-00440-t001:** Primer sequence for amplification of F-gene.

Name	Sequence	Product, bp
NDV VII_F1	ATG ATC CAT CTC GAC TGC T (4448)	823
NDV VII_R1	GTC AGC TGA GTT AAT GCA G (5250)	
NDV VII_F2	TAT CAC AAC TAT CAG TGG CA (5064)	629
NDV VII_R2	CGT AGT GAG TGC GCC TTC AGT (5693)	
NDV VII_F3	AGA AGA GCT TGA CAC CTC ATA CT (5539)	738
NDV VII_R3	ACT TGA CAG GTT ATC GGA ATT G (6277)	

## Data Availability

Data is contained within the article or Appendix A.

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
