# Peer review of "Phylogenetic Analysis of a Newcastle Disease Virus Strain Isolated from Domestic Poultry and Its Potential for Vaccine Development in the Republic of Kazakhstan"

_vaccines, 2025, doi:10.3390/vaccines13050440_

Round 1
Reviewer 1 Report
Comments and Suggestions for Authors
The manuscript presents the molecular characterisation of the Newcastle Disease Virus (NDV) circulating in Kazakhstan and the development of an inactivated vaccine using this circulating strain with two different adjuvants: Montanide ISA 70 VG and Montanide ISA 78 VG. The manuscript is generally well written, but several improvements are needed to make it more informative for the reader.
Main comments:
- Although it is mentioned that supporting information can be downloaded, no supplementary material is currently available. I think the reader would benefit if the authors could provide a supplementary table with detailed information on individual clinical observations seen after challenge with the virulent NDV strain. The inclusion of images showing lesions in the control group after challenge would further strengthen the manuscript.
- A supplementary table showing individual haemagglutination inhibition (HI) test results should also be included. Results obtained at 7 days post-vaccination (PV) are missing from Figure 2 and should be provided.
- Authors should use standardised dosage terminology throughout the manuscript, e.g. 10⁵ EID₅₀/ml instead of 100,000 EID₅₀/cm³ or 0.5 ml instead of 0.5 cm³.
- Detailed information on the chickens and eggs used in the experiments is missing. It should be specified whether specific pathogen-free (SPF) chicks or chickens with MDA for NDV were used and the source of the animals and eggs.
Specific comments:
- Line 40: The reference to OIE (2012) should be updated as Chapter 3.3.10 has been revised in 2021.
- Line 42: Reference 1 should be fully explained or updated.
- Lines 57-65: References are missing and should be added.
- Line 134: More details of the technique used should be provided.
- Line 137: A brief description of the procedure should be included rather than simply stating that it was performed according to the manufacturer's instructions.
Line 180: Clarification is needed as to why the intramuscular (i/m) method was used for the challenge test.
- References: The reference list needs to be thoroughly reviewed and updated as necessary.
Author Response
Dear Reviewer,
We sincerely appreciate your valuable comments and suggestions, which have contributed to improving our article. We have carefully considered each of your remarks and made the necessary revisions accordingly.
Main comments1: Although it is mentioned that supporting information can be downloaded, no supplementary material is currently available. I think the reader would benefit if the authors could provide a supplementary table with detailed information on individual clinical observations seen after challenge with the virulent NDV strain. The inclusion of images showing lesions in the control group after challenge would further strengthen the manuscript.
Response 1: We have prepared an additional table with detailed data on clinical observations recorded after infection with the virulent NDV strain. The table is included as supplementary material 1. Images of diseased birds were of poor quality; therefore, we did not include them in the article or the supplementary material.
Main comments 2: A supplementary table showing individual haemagglutination inhibition (HI) test results should also be included. Results obtained at 7 days post-vaccination (PV) are missing from Figure 2 and should be provided.
Response 2.We have included a table with the results of individual HI tests as Supplementary Material 2.
Initially, we did not include data from day 7 post-vaccination in the figure, as all results were negative, making the figure uninformative. However, we have now updated Figure 2 in accordance with your comment. The final decision regarding the format of figure 2 is left to the editorial team. Page number 7, paragraph 3.5, line 259.
Main comments 3: Authors should use standardised dosage terminology throughout the manuscript, e.g. 10⁵ EID₅₀/ml instead of 100,000 EID₅₀/cm³ or 0.5 ml instead of 0.5 cm³
Response 3. We have standardized all dosage terminology as follows:
- Used the notation 10⁵ EID₅₀/mL.
- Adjusted the volume to 0.5 mL instead of 0.5 cm³, line 20, line 182, line 285
Main comments 4 Detailed information on the chickens and eggs used in the experiments is missing. It should be specified whether specific pathogen-free (SPF) chicks or chickens with MDA for NDV were used and the source of the animals and eggs:
Response 4: SPF chicks were used in the experiments.Source of animals and eggs: SPF eggs and Super Nick breed chicks hatched from SPF eggs obtained from the Podmoklov nursery (Moscow, Russia). This information has been added to the manuscript. Page number 3, paragraph 2.1., line 85-88.
Specific comments 1: Line 40: The reference to OIE (2012) should be updated as Chapter 3.3.10 has been revised in 2021
Response to specific comment 1: The reference has been updated to the latest version (2021). line 31.
Specific comments 2: Line 42: Reference 1 should be fully explained or updated.
Response to specific comment 2: Reference 1 has been updated and included in the "References" section. Page 12, line 418.
Specific comments 3: Lines 57-65: References are missing and should be added.
Response to specific comment 3: The missing references have been added:
- Cornax I., Miller P.J., Afonso C.L., et al.
- Winterfield R.W., Dhillon A.S., Alby L.J., et al.
- Kim S-H., Chen Z., Yoshida A., et al
Page number 2, paragraph 1., line 51-57 and page number12, line 436-442.
Specific comments 4: -Line 134: More details of the technique used should be provided.
Response to specific comment 4: The description has been added, including the stages of vaccine preparation using Montanide ISA 70 VG and 78 VG. Page number 4, paragraph 2.3., line 130-138
Specific comments 5:-Line 137: A brief description of the procedure should be included rather than simply stating that it was performed according to the manufacturer's instructions.
Response to specific comment 5: The procedure description has been added, along with a brief rationale for vaccine preparation. Page number 4, paragraph 2.3., line 130-138
Specific comments 6: Line 180: Clarification is needed as to why the intramuscular (i/m) method was used for the challenge test.
Response to specific comment 6: The intramuscular (IM) method was chosen for the challenge test as it ensures precise and controlled administration of the material, which is critical for the reliability of the results. This method minimizes material loss and provides the necessary uniform distribution, which is particularly important for this study.
Additionally, aerosol vaccination was not an option due to the lack of appropriate equipment. It is also worth noting that in our previously published studies, we thoroughly examined the method for determining the challenge dose. Based on our findings, intramuscular administration of the virulent strain has demonstrated reliability and accuracy, making it the preferred method for this experiment. Available online: http://iramn.ru/journals/bbm/2024/12/8495/
Specific comments 7: References: The reference list needs to be thoroughly reviewed and updated as necessary.
Response to specific comment 7: The reference list has been fully reviewed and updated. All references have been formatted in accordance with the required standards.
Reviewer 2 Report
Comments and Suggestions for Authors
To Editor, Authors The manuscript “Phylogenetic analysis of Newcastle disease virus and its potential in vaccine development”, Vaccines Journal is of interest for virologists, especially for veterinary virology researchers: for epidemiologists of NDV and veterinarian specialists. The authors suggest original strain for vaccine producing. The discussion section seems to me very interesting and meaningful. But still has some questions to be addressed. General Comments: 1 Since the Title sounds very general - about the phylogenetic analysis of the Newcastle disease virus, consider specifying any details, for example, that it was isolated in Kazakhstan from poultry or something else. In its current form, the title sounds like it's for a review article. 2 The final part of the Introduction section lacks the specific purpose/aim of this study, please consider 3 In my opinion, the organization of the Materials and Methods Section needs to be reviewed. Perhaps, before describing sequencing, it would be worthwhile to include information about the viral strain "PMV-1/Acana/chicken/49/98", maybe as a subsection. Where is he from? is it from a Strain collection of the Instituteor field research? Because it is not clear where it came from to the study and you firstly describe the methods of its sequencing and analysis. The strain appears only in the current department of methods for the preparation of the vaccine. 4 In section “3.5. Analysis of the immunogenicity assessment of inactivated emulsified vaccines” you should indicate that in control group you did not find antibodies if so. 5 Figure 3 shows the incorrect designations G1 and G2, instead of G3 and G4, as they should be according to the description in the text and the materials and methods section. Also, the graph line of the control group should probably end on day 5, when all the birds died. Please check all designations on all Figures. Specific Comments: Line 27: Please specify that they all vaccinated survived Line 30: What is “the epizootic topical strain” you mentioned? Is it “typical”? Lines 43-44: Please indicate the current taxonomy according to ICTV (species Orthoavulavirus javaense, genus Orthoavulavirus). Line 47: The names commonly used to designate viral polymerase are: large protein, large polymerase protein, L protein, RNA-dependent RNA polymerase. Please choose one name. The combination "large RNA-dependent polymerase" is not very good.
Comments on the Quality of English LanguageI do not feel sufficiently qualified to evaluate the English language aspects of this Manuscript
Author Response
Dear Reviwer,
We appreciate your comments on the manuscript "Phylogenetic Analysis of the Newcastle Disease Virus and Its Potential for Vaccine Development." We have carefully considered the feedback and made the necessary revisions.
General comments 1: Since the Title sounds very general - about the phylogenetic analysis of the Newcastle disease virus, consider specifying any details, for example, that it was isolated in Kazakhstan from poultry or something else. In its current form, the title sounds like it's for a review article.
Response 1: We have revised the manuscript title and hope that the new title eliminates any impression of a review article.
Updated title: "Phylogenetic analysis of a Newcastle disease virus strain isolated from domestic poultry and its potential for vaccine development in the Republic of Kazakhstan." Page 1.
General comments 2: The final part of the Introduction section lacks the specific purpose/aim of this study, please consider
Response 2: The concluding section of the "Introduction" has been revised, and the research objective has been explicitly stated. Page number 2, paragraph 1., line 60-62
General comments 3: In my opinion, the organization of the Materials and Methods Section needs to be reviewed. Perhaps, before describing sequencing, it would be worthwhile to include information about the viral strain "PMV-1/Acana/chicken/49/98", maybe as a subsection. Where is he from? is it from a Strain collection of the Instituteor field research? Because it is not clear where it came from to the study and you firstly describe the methods of its sequencing and analysis. The strain appears only in the current department of methods for the preparation of the vaccine.
Response 3: We have revised the "Materials and Methods" section by adding information about the PMV-1/Astana/chicken/49/98 strain to subsection paragraph 2.1 . The following text has been added to subsection "2.1 Sequencing of the F Gene of NDV":
"For the research study, the Newcastle disease virus strain PMV-1/Astana/chicken/49/98 was used. This strain was isolated from domestic poultry in Kazakhstan and maintained in the microbial collection laboratory of the Research Institute for Biological Safety Problems (RIBSP), Kazakhstan." Page number 2, line 80-83
General comments 4: In section “3.5. Analysis of the immunogenicity assessment of inactivated emulsified vaccines” you should indicate that in control group you did not find antibodies if so.
Response 4: The text has been updated to include information stating that no antibodies were detected in the control group. The revised text now reads:
"In contrast, no antibodies were detected in the control group either before the study or during the assessment of vaccine immunogenicity." Page number 8, line 274
General comments 5: Figure 3 shows the incorrect designations G1 and G2, instead of G3 and G4, as they should be according to the description in the text and the materials and methods section. Also, the graph line of the control group should probably end on day 5, when all the birds died. Please check all designations on all Figures.
Response 5: Incorrect labels G1 and G2 have been replaced with G3 and G4, as specified in the text and the "Materials and Methods" section. The control group’s graph line has been adjusted to end on day 6, when all birds perished. Additionally, all figure labels have been reviewed, and any identified inconsistencies have been corrected. Page number 8, line 280
Specific comments 1: Line 27: Please specify that they all vaccinated survived
Response to specific comments 1 : Information has been added to confirm that all vaccinated birds survived. The revised text now reads: "Throughout the study, all vaccinated birds remained alive until the end of the experiment, demonstrating the high efficacy of inactivated vaccines based on the PMV-1/Astana/chicken/49/98 strain with different adjuvants against Newcastle disease after a single vaccination." Page number 2, paragraph 2.1, line 80-84
Specific comments 2: Line 30: What is “the epizootic topical strain” you mentioned? Is it “typical”?
Response to specific comments 2 : We clarify that the study refers to the local strain PMV-1/Astana/chicken/49/98, isolated in the Republic of Kazakhstan. Corrections have been made, replacing the word "topical" with "local."
Specific comments 3: Lines 43-44: Please indicate the current taxonomy according to ICTV (species Orthoavulavirus javaense, genus Orthoavulavirus).
Response to specific comments 3 : Yes, we have made the necessary corrections to the text. The updated version now reads:"Newcastle disease virus (NDV), according to ICTV taxonomy, belongs to the species Orthoavulavirus javaense within the genus Orthoavulavirus. It is an enveloped, negative-sense, single-stranded RNA virus." Page number 1, paragraph 1, line 32-34
Specific comments 4: Line 47: The names commonly used to designate viral polymerase are: large protein, large polymerase protein, L protein, RNA-dependent RNA polymerase. Please choose one name. The combination "large RNA-dependent polymerase" is not very good.
Response to specific comments 4 : We have reviewed the terminology and replaced "large RNA-dependent polymerase" with "large polymerase protein", which is the commonly accepted term for the L protein. We hope that these revisions address your comments satisfactorily. Page number 2, paragraph 1, line 37